# VICReg: Variance-Invariance-Covariance Regularization for Self-Supervised Learning

## Abstract

Recent self-supervised methods for image representation learning maximize the agreement between embedding vectors produced by encoders fed with different views of the same image. The main challenge is to prevent a *collapse* in which the encoders produce constant or non-informative vectors. We introduce VICReg (Variance-Invariance-Covariance Regularization), a method that explicitly avoids the collapse problem with two regularizations terms applied to both embeddings separately: (1) a term that maintains the variance of each embedding dimension above a threshold, (2) a term that decorrelates each pair of variables. Unlike most other approaches to the same problem, VICReg does *not* require techniques such as: weight sharing between the branches, batch normalization, feature-wise normalization, output quantization, stop gradient, memory banks, etc., and achieves results on par with the state of the art on several downstream tasks. In addition, we show that our variance regularization term stabilizes the training of other methods and leads to performance improvements.

## 1  Introduction

Self-supervised representation learning has made significant progress over the last years, almost reaching the performance of supervised baselines on many downstream tasks [1, 2, 3, 4, 5, 6, 7, 8, 9]. Several recent approaches rely on a *joint embedding architecture* in which two networks are trained to produce similar embeddings for different views of the same image. A popular instance is the Siamese network architecture [10], where the two networks share the same weights. The main challenge with joint embedding architectures is to prevent a *collapse* in which the two branches ignore the inputs and produce identical output vectors. There are two main approaches to preventing collapse: contrastive methods and information maximization methods. Contrastive methods [3, 11, 12] use a loss that explicitly pushes the embeddings of dissimilar images away from each other. They often require a mining procedure to search for offending dissimilar samples from a memory bank [3] or from the current batch [12]. Contrastive methods tend to be costly, require large batch sizes or memory banks, and do not seem to scale well with the dimension of the embedding. Quantization-based approaches [5, 13] force the embeddings of different samples to belong to different clusters on the unit sphere. Collapse is prevented by ensuring that the assignment of samples to clusters is as uniform as possible. A similarity term encourages the cluster assignment score vectors from the two branches to be similar. More recently, a few methods have appeared that do not rely on contrastive samples or vector quantization, yet produce high-quality representations, for example BYOL [6] and SimSiam [7]. They exploit several tricks: batch-wise or feature-wise normalization, a "momentum encoder" in which the parameter vector of one branch is a low-pass-filtered version of the parameter vector of the other branch [6, 14], or a stop-gradient operation in one of the branches [7]. The dynamics of learning in these methods, and how they avoid collapse, is not fully understood, although theoretical and empirical studies point to the crucial importance of batch-wise or feature-wise normalization [14, 15]. Finally, an alternative class of collapse prevention methods relies on maximizing the information content of the embedding [9, 16]. These methods prevent *informational collapse* by decorrelating

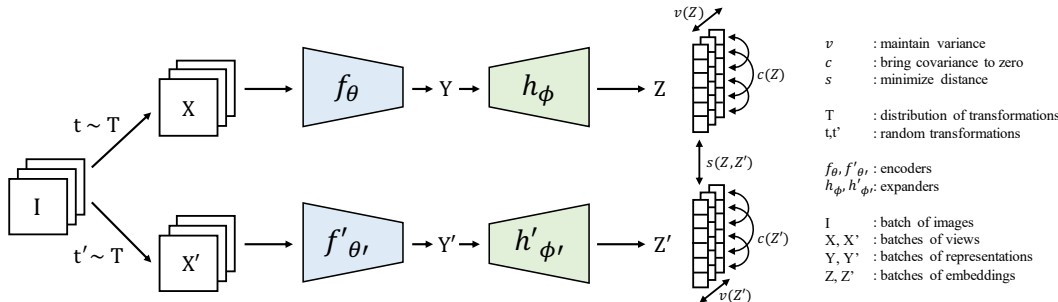

Figure 1: **VICReg: joint embedding architecture with variance, invariance and covariance regularization.** Given a batch of images $I$, two batches of different views $X$ and $X'$ are produced and are then encoded into representations $Y$ and $Y'$. The representations are fed to an expander producing the embeddings $Z$ and $Z'$. The distance between two embeddings from the same image is minimized, the variance of each embedding variable over a batch is maintained above a threshold, and the covariance between pairs of embedding variables over a batch are attracted to zero, decorrelating the variables from each other. Although the two branches do not require identical architectures nor share weights, in most of our experiments, they are Siamese with shared weights: the encoders are ResNet-50 backbones with output dimension 2048. The expanders have 3 fully-connected layers of size 8192.

every pair of variables of the embedding vectors. This indirectly maximizes the information content of the embedding vectors. The Barlow Twins method drives the normalized cross-correlation matrix of the two embeddings towards the identity [9], while the Whitening-MSE method whitens and spreads out the embedding vectors on the unit sphere [16].

## 2 VICReg: intuition

We introduce VICReg (Variance-Invariance-Covariance Regularization), a self-supervised method for training joint embedding architectures based on the principle of preserving the information content of the embeddings. The basic idea is to use a loss function with three terms:

- **Invariance**: the mean square distance between the embedding vectors.
- **Variance**: a hinge loss to maintain the standard deviation (over a batch) of each variable of the embedding above a given threshold. This term forces the embedding vectors of samples within a batch to be different.
- **Covariance**: a term that attracts the covariances (over a batch) between every pair of (centered) embedding variables towards zero. This term decorrelates the variables of each embedding and prevents an *informational collapse* in which the variables would vary together or be highly correlated.

Variance and Covariance terms are applied to both branches of the architecture separately, thereby preserving the information content of each embedding at a certain level and preventing informational collapse independently for the two branches. The main contribution of this paper is the Variance preservation term, which explicitly prevents a collapse due to a shrinkage of the embedding vectors towards zero. The Covariance criterion is borrowed from the Barlow Twins method and prevents informational collapse due to redundancy between the embedding variables [9]. VICReg is more generally applicable than most of the aforementioned methods because of fewer constraints on the architecture. In particular, VICReg:

- does not require that the weights of the two branches be shared, not that the architectures be identical, nor that the inputs be of the same nature;
- does not require a memory bank, nor contrastive samples, nor a large batch size;
- does not require batch-wise nor feature-wise normalization; and
- does not require vector quantization nor a predictor module.

Other methods require asymmetric stop gradient operations, as in SimSiam [7], weight sharing between the two branches as in classical Siamese nets, or weight sharing through exponential moving

average dampening with stop gradient in one branch, as in BYOL and MoCo [3, 6, 17], large batches of contrastive samples, as in SimCLR [12], or batch-wise and/or feature-wise normalization [5, 6, 7, 9, 16]. One of the most interesting feature of VICReg is the fact that the two branches are not required to share the same parameters, architecture, or input modality. This opens the door to the use of non-contrastive self-supervised joint-embedding for multi-modal signals, such as video and audio. We demonstrate the effectiveness of the proposed approach by evaluating the representations learned with VICReg on several downstream image recognition tasks including linear head and semi-supervised evaluation protocols for image classification on ImageNet [18], and other classification, detection and instance segmentation tasks. Furthermore, we show that incorporating variance preservation into other self-supervised joint-embedding methods yields better training stability and performance improvement on downstream tasks. More generally, we show that VICReg is an explicit and effective, yet simple method for preventing collapse in self-supervised joint-embedding learning.

## 3 Related work

**Contrastive learning.** In contrastive SSL methods applied to joint embedding architectures, the output embeddings for a sample and its distorted version are brought close to each other, while other samples and their distortions are pushed away. The method is most often applied to Siamese architectures in which the two branches have identical architectures and share weights [2, 3, 10, 11, 12, 17, 19, 20, 21, 22, 23]. Many authors use the InfoNCE loss [22] in which the repulsive force is larger for contrastive samples that are closer to the reference. While these methods yield good performance, they require large amounts of contrastive pairs in order to work well. These contrastive pairs can be sampled from a memory bank as in MoCo [3], or given by the current batch of data as in SimCLR [12], with a significant memory footprint. This downside of contrastive methods motivates a search for alternatives.

**Clustering methods.** Instead of viewing each sample as its own class, clustering-based methods group them into clusters based on some similarity measure [5, 13, 24, 25, 26, 27, 28, 29, 30, 31]. DeepCluster [13] uses $k$-means assignments of representations from previous iterations as pseudo-labels for the new representations, which requires an expensive clustering phase done asynchronously, and makes the method hard to scale up. SwAV [5] mitigates this issue by learning the clusters online while maintaining a balanced partition of the assignments through the Sinkhorn-Knopp transform [32]. These clustering approaches can be viewed as contrastive learning at the level of clusters which still requires a lot of negative comparisons to work well.

**Distillation methods.** Recent proposals such as BYOL, SimSiam, OBoW and variants [6, 7, 8, 14, 33] have shown that collapse can be avoided by using architectural tricks inspired by knowledge distillation [34]. These methods train a student network to predict the representations of a teacher network, for which the weights are a running average of the student network's weights [6], or are shared with the student network, but no gradient is back-propagated through the teacher [7]. These methods are effective, but there is no clear understanding of how t but suffer from a lack of explainability regarding the way collapsing solutions are avoided. Alternatively, the images can be represented as bags of word over a dictionary of visual features, which effectively prevents collapse. In [33] and [8] the dictionary is obtained by off-line or on-line clustering. By contrast, our method explicitly prevents collapse in the two branches independently, which removes the requirement for shared weights and identical architecture, opening the door to the application of joint-embedding SSL to multi-modal signals.

**Information maximization methods.** A principle to prevent collapse is to maximize the information content of the embeddings. Two such methods were recently proposed: W-MSE [16] and Barlow Twins [9]. In W-MSE, an extra module transforms the embeddings into the eigenspace of their covariance matrix (whitening or Karhunen-Loève transform), and forces the vectors thereby obtained to be uniformly distributed on the unit sphere. In Barlow Twins, a loss term attempts to make the normalized cross-correlation matrix of the embedding vectors from the two branches to be close to the identity. Both methods attempt to produce embedding variables that are decorrelated from each other, thus preventing an *informational collapse* in which the variables carry redundant information. Because all variables are normalized over a batch, there is no incentive for them to shrink nor expand. This seems to sufficient to prevent collapse. Our method borrows the decorrelation mechanism of Barlow Twins. But it includes an explicit variance-preservation term for each variable of the two embeddings and thus does not require any normalization.

# 4 VICReg: detailed description

VICReg follows recent trends in self-supervised learning [5, 6, 7, 9, 12] and is based on a *joint embedding architecture*. Contrary to many previous approaches, our architecture may be completely symmetric or completely asymmetric with no shared structure or parameters between the two branches. In most of our experiments, we use a Siamese net architecture in which the two branches are identical and share weights. Each branch consists of an *encoder* $f_\theta$ that outputs the representations (used for downstream tasks), followed by an *expander* $h_\phi$ that maps the representations into an embedding space where the loss function will be computed. The role of the expander is twofold: (1) eliminate the information by which the two representations differ, (2) expand the dimension in a non-linear fashion so that decorrelating the embedding variables will reduce the dependencies (not just the correlations) between the variables of the representation vector. The loss function uses a term $s$ that learns invariance to data transformations and is regularized with a variance term $v$ that prevents norm collapse and a covariance term $c$ that prevents informational collapse by decorrelating the different dimensions of the vectors. After pretraining, the expander is discarded and the representations of the encoder are used for downstream tasks.

## 4.1 Method

Given an image $i$ sampled from a dataset $\mathcal{D}$, two transformations $t$ and $t'$ are sampled from a distribution $\mathcal{T}$ to produce two different views $x = t(i)$ and $x' = t'(i)$ of $i$. These transformations are random crops of the image, followed by color distortions. The distribution $\mathcal{T}$ is described in Appendix C. The views $x$ and $x'$ are first encoded by $f_\theta$ into their *representations* $y = f_\theta(x)$ and $y' = f_\theta(x')$, which are then mapped by the expander $h_\phi$ onto the *embeddings* $z = h_\phi(y)$ and $z' = h_\phi(y')$. The loss is computed at the embedding level on $z$ and $z'$.

We describe here the variance, invariance and covariance terms that compose our loss function. The images are processed in batches, and we denote $Z = [z_1, \ldots, z_n]$ and $Z' = [z'_1, \ldots, z'_n]$ the two batches composed of $n$ vectors of dimension $d$, of embeddings coming out of the two branches of the siamese architecture. We denote by $z^j$ the vector composed of each value at dimension $j$ in all vectors in $Z$. We define the variance regularization term $v$ as a hinge function on the standard deviation of the embeddings along the batch dimension:

$$v(Z) = \frac{1}{d} \sum_{j=1}^{d} \max(0, \gamma - S(z^j, \epsilon)), \tag{1}$$

where $S$ is the regularized standard deviation defined by:

$$S(x, \epsilon) = \sqrt{\text{Var}(x) + \epsilon}, \tag{2}$$

$\gamma$ is a constant target value for the standard deviation, fixed to $1$ in our experiments, $\epsilon$ is a small scalar preventing numerical instabilities. This criterion encourages the variance inside the current batch to be equal to $\gamma$ along each dimension, preventing collapse with all the inputs mapped on the same vector. Using the standard deviation and not directly the variance is crucial. Indeed, if we take $S(x) = \text{Var}(x)$ in the hinge function, the gradient of $S$ with respect to $x$ becomes close to $0$ when $x$ is close to $\bar{x}$. In this case, the gradient of $v$ also becomes close to $0$ and the embeddings collapse. We define the covariance matrix of $Z$ as:

$$C(Z) = \frac{1}{n-1} \sum_{i=1}^{n} (z_i - \bar{z})(z_i - \bar{z})^T, \quad \text{where} \quad \bar{z} = \frac{1}{n} \sum_{i=1}^{n} z_i. \tag{3}$$

Inspired by Barlow Twins [9], we can then define the covariance regularization term $c$ as the sum of the squared off-diagonal coefficients of $C(Z)$, with a factor $1/d$ that scales the criterion as a function of the dimension:

$$c(Z) = \frac{1}{d} \sum_{i \neq j} [C(Z)]_{i,j}^2. \tag{4}$$

This term encourages the off-diagonal coefficients of $C(Z)$ to be close to $0$, decorrelating the different dimensions of the embeddings and preventing them from encoding similar information. Decorrelation at the embedding level ultimately has a decorrelation effect at the representation level, which is a non trivial phenomenon that we study in Appendix D. We finally define the invariance criterion $s$

between $Z$ and $Z'$ as the mean-squared euclidean distance between each pair of vectors, without any normalization:

$$s(Z, Z') = \frac{1}{n} \sum_i \|z_i - z_i'\|_2^2. \tag{5}$$

The overall loss function is a weighted average of the invariance, variance and covariance terms:

$$\ell(Z, Z') = \lambda s(Z, Z') + \mu[v(Z) + v(Z')] + \nu[c(Z) + c(Z')], \tag{6}$$

where $\lambda$, $\mu$ and $\nu$ are hyper-parameters controlling the importance of each term in the loss. In our experiments, we set $\nu = 1$ and perform a grid search on the values of $\lambda$ and $\mu$ with the base condition $\lambda = \mu > 1$. The overall objective function taken on all images over an unlabelled dataset $\mathcal{D}$ is given by:

$$\mathcal{L} = \sum_{I \in \mathcal{D}} \sum_{t, t' \sim \mathcal{T}} \ell(Z^I, Z'^I), \tag{7}$$

where $Z^I$ and $Z'^I$ are the batches of embeddings corresponding to the batch of images $I$ transformed by $t$ and $t'$. The objective is minimized for several epochs, over the encoder parameters $\theta$ and expander parameters $\phi$. We illustrate the architecture and loss function of VICReg in Figure 1.

## 4.2 Implementation details

Implementation details for pretraining with VICReg on the 1000-classes ImagetNet[1] dataset without labels are as follows. Coefficients $\lambda$ and $\mu$ are 25 and $\nu$ is 1 in Eq. (6), and $\epsilon$ is 0.0001 in Eq. (1). The encoder network $f_\theta$ is a standard ResNet-50 backbone [35] with 2048 output units. The expander $h_\phi$ is composed of two fully-connected layers with batch normalization (BN) [36] and ReLU, and a third linear layer. The sizes of all 3 layers were set to 8192. As with Barlow Twins, performance improves when the size of the expander layers is larger than the dimension of the representation. The impact of the expander dimension on performance is studied in Appendix D. The training protocol follows those of BYOL and Barlow Twins: LARS optimizer [37, 38] run for 1000 epochs with a

**Algorithm 1:** VICReg pseudocode.

```
# f: encoder network, lambda, mu, nu: coefficients of the invariance, variance and
       covariance losses, N: batch size, D: dimension of the representations
# mse_loss: Mean square error loss function, off_diagonal: off-diagonal elements
       of a matrix, relu: ReLU activation function

for x in loader: # load a batch with N samples
    # two randomly augmented versions of x
    x_a, x_b = augment(x)

    # compute representations
    z_a = f(x_a) # N x D
    z_b = f(x_b) # N x D

    # invariance loss
    sim_loss = mse_loss(z_a, z_b)

    # variance loss
    std_z_a = torch.sqrt(z_a.var(dim=0) + 1e-04)
    std_z_b = torch.sqrt(z_b.var(dim=0) + 1e-04)
    std_loss = torch.mean(relu(1 - std_z_a)) + torch.mean(relu(1 - std_z_b))

    # covariance loss
    z_a = z_a - z_a.mean(dim=0)
    z_b = z_b - z_b.mean(dim=0)
    cov_z_a = (z_a.T @ z_a) / (N - 1)
    cov_z_b = (z_b.T @ z_b) / (N - 1)
    cov_loss = off_diagonal(cov_z_a).pow_(2).sum() / D
             + off_diagonal(cov_z_b).pow_(2).sum() / D

    # loss
    loss = lambda * sim_loss + mu * std_loss + nu * cov_loss

    # optimization step
    loss.backward()
    optimizer.step()
```

weight decay of $10^{-6}$ and a learning rate $lr = batch\_size/256 \times base\_lr$, where $batch\_size$ is set to 2048 by default and $base\_lr$ is a base learning rate set to 0.2. The learning rate follows a cosine decay schedule [39], starting from 0 with 10 warmup epochs and with final value of 0.002.

## 5 Results

In this section, we evaluate the representations obtained after self-supervised pretraining of a ResNet-50 [35] backbone with VICReg during 1000 epochs, on the training set of ImageNet, using the training protocol described in section 4.

## 5.1 Evaluation on ImageNet

Following the ImageNet [18] linear evaluation protocol, we train a linear classifier on top of the frozen representations of the ResNet-50 backbone pretrained with VICReg. We also evaluate the performance of the backbone when fine-tuned with a linear classifier on a subset of ImageNet's training set using 1% or 10% of the labels, using the split of [12]. We give implementation details about the optimization procedure for these tasks in Appendix C. We have applied the training procedure described in section 4 with three different random initialization. The numbers reported in Table 1 for VICReg are the mean scores, and we have observed that the difference between worse and best run is lower than 0.1% accuracy for linear classification, which shows that VICReg is a

---

[1]ImageNet is free to use for research purpose and non-commercial use only.

Table 1: **Evaluation on ImageNet.** Evaluation of the representations obtained with a ResNet-50 backbone pretrained with VICReg on: (1) linear classification on top of the frozen representations from ImageNet; (2) semi-supervised classification on top of the fine-tuned representations from 1% and 10% of ImageNet samples. We report Top-1 and Top-5 accuracies (in %). Top-3 best self-supervised methods are underlined.

| Method | Linear Classification | | Semi-supervised Classification | | | |
| | Top-1 | Top-5 | Top-1 | | Top-5 | |
| | | | 1% | 10% | 1% | 10% |
|---|---|---|---|---|---|---|
| Supervised | 76.5 | - | 25.4 | 56.4 | 48.4 | 80.4 |
| MoCo [3] | 60.6 | - | - | - | - | - |
| PIRL [2] | 63.6 | - | - | - | 57.2 | 83.8 |
| CPC v2 [40] | 63.8 | - | - | - | - | - |
| CMC [41] | 66.2 | - | - | - | - | - |
| SimCLR [12] | 69.3 | 89.0 | 48.3 | 65.6 | 75.5 | 87.8 |
| MoCo v2 [17] | 71.1 | - | - | - | - | - |
| SimSiam [7] | 71.3 | - | - | - | - | - |
| SwAV [5] | 71.8 | - | - | - | - | - |
| InfoMin Aug [4] | 73.0 | 91.1 | - | - | - | - |
| OBoW [8] | 73.8 | - | - | - | 82.9 | 90.7 |
| BYOL [6] | 74.3 | 91.6 | 53.2 | 68.8 | 78.4 | 89.0 |
| SwAV (w/ multi-crop) [5] | 75.3 | - | 53.9 | 70.2 | 78.5 | 89.9 |
| Barlow Twins [9] | 73.2 | 91.0 | 55.0 | 69.7 | 79.2 | 89.3 |
| VICReg (ours) | 73.2 | 91.1 | 54.8 | 69.5 | 79.4 | 89.5 |

very stable algorithm. Lack of time has prevented us from doing the same for the semi-supervised classification experiments, and the experiments of section 5.2 and 6, but we expect similar conclusion to hold. We compare in Table 1 our results on both tasks against other methods on the validation set of ImageNet. The performance of VICReg is on par with the state of the art without using the negative pairs of SimCLR, the clusters of SwAV, the bag-of-words representations of OBoW, or any asymmetric networks architectural tricks such as the momentum encoder of BYOL and the stop-gradient operation of SimSiam. The performance is comparable to that of Barlow Twins, which shows that VICReg's more explicit way of constraining the variance and comparing views has the same power than maximizing cross-correlations between pairs of twin dimensions. The main advantage of VICReg is the modularity of its objective function and the potential applicability to multi-modal setups.

## 5.2 Transfer to other downstream tasks

Following the setup from [2], we train a linear classifier on top of the frozen representations learnt by our pretrained ResNet-50 backbone on a variety of different datasets: the Places205 [42] scene classification dataset, the VOC07 [43] multi-label image classification dataset and the iNaturalist2018 [44] fine-grained image classification dataset[2]. We then evaluate the quality of the representations by transferring to other vision tasks including VOC07+12 [43] object detection using Faster R-CNN [45] with a R50-C4 backbone, and COCO [46] instance segmentation using Mask-R-CNN [47] with a R50-FPN backbone. We give implementation details in Appendix C. We report the performance in Table 2, VICReg performs on par with most concurrent methods, and better than Barlow Twins, across all classification tasks, but is slightly behind the top-3 on detection tasks. This could be explained by the fact that VICReg learns representations that are more invariant to transformation, but eliminates more low-level information about the images than the other methods.

## 6 Ablation study

In this section we study how the different components of our method contribute to its performance, as well as how they interact with components from other self-supervised methods. All reported

---

[2] Places205 was released under the CC-BY license. Pascal VOC, iNaturalist18 and COCO are free to use for research purposes and non-commercial use only.

Table 2: **Transfer learning on downstream tasks.** Evaluation of the representations from a ResNet-50 backbone pretrained with VICReg on: (1) linear classification tasks on top of frozen representations, we report Top-1 accuracy (in %) for Places205 [42] and iNat18 [44], and mAP for VOC07 [43]; (2) object detection with fine-tunning, we report $AP_{50}$ for VOC07+12 using Faster R-CNN with C4 backbone [45]; (3) object detection and instance segmentation, we report AP for COCO [46] using Mask R-CNN with FPN backbone [47]. We use † to denote the experiments run by us. Top-3 best self-supervised methods are underlined.

| | Linear Classification | | | Object Detection | | |
|---|---|---|---|---|---|---|
| Method | Places205 | VOC07 | iNat18 | VOC07+12 | COCO det | COCO seg |
| Supervised | 53.2 | 87.5 | 46.7 | 81.3 | 39.0 | 35.4 |
| MoCo [3] | 46.9 | 79.8 | 31.5 | - | - | - |
| PIRL [2] | 49.8 | 81.1 | 34.1 | - | - | - |
| SimCLR [12] | 52.5 | 85.5 | 37.2 | - | - | - |
| MoCo v2 [17] | 51.8 | 86.4 | 38.6 | 82.5 | 39.8 | 36.1 |
| SimSiam [7] | - | - | - | 82.4 | - | - |
| BYOL [6] | 54.0 | 86.6 | 47.6 | - | 40.4† | 37.0† |
| SwAV (w/ multi-crop) [5] | 56.7 | 88.9 | 48.6 | 82.6 | 41.6 | 37.8 |
| OBoW [8] | 56.8 | 89.3 | - | 82.9 | - | - |
| Barlow Twins [6] | 54.1 | 86.2 | 46.5 | 82.6 | 40.0† | 36.7† |
| VICReg (ours) | 54.3 | 86.6 | 47.0 | 82.4 | 39.4 | 36.4 |

results are obtained on the linear evaluation protocol using a ResNet-50 backbone and 100 epochs of pretraining, which gives results consistent with those obtained with 1000 epochs of pretraining. The optimization setting used for each experiment is described in Appendix C.

**Asymmetric networks.** We study the impact of different components used in asymmetric architectures and the effects of adding variance and covariance regularization, in terms of performance and training stability. Starting from a simple symmetric architecture with an encoder and an expander without batch normalization, which correspond to VICReg without batch normalization in the expander, we progressively add batch normalization in the inner layers of the expander, a predictor, a stop-gradient operation and a momentum encoder. We use the training protocol and architecture of SimSiam [7] when a stop-gradient is used and the training protocol and architecture of BYOL [6] when a momentum encoder is used. The predictor as used in SimSiam and BYOL is a learnable module $g_\psi$ that predicts the embedding of a view given the embedding of the other view of the same image. If $z$ and $z'$ are the embeddings of two views of an image, then $p = g_\psi(z)$ and $p' = g_\psi(z')$ are the predictions of each view. The invariance loss function of Eq. (5) is now computed between a batch of embeddings $Z = [z_1, \ldots, z_n]$ and the corresponding batch of predictions $P = [p'_1, \ldots, p'_n]$, then symmetrized:

$$s(Z, Z', P, P') = \frac{1}{2n} \sum_i D(z_i - p'_i) + \frac{1}{2n} \sum_i D(z'_i - p_i), \qquad (8)$$

where $D$ is a distance function that depends on the method used. BYOL uses the mean square error between $l_2$-normalized vectors, SimSiam uses the negative cosine similarity loss and VICReg uses the mean square error without $l_2$-normalization. The variance and covariance terms are regularizing the output $Z$ and $Z'$ of the expander, which we empirically found to work better than regularizing the output of the predictor. We compare different settings in Table 3, based on the default data augmentation, optimization and architecture settings of the original BYOL, SimSiam and VICReg methods. In all settings, the absence of BN indicates that BN is also removed in the predictor when one is used.

We analyse first the impact of variance regularization (VR) in the different settings. When using VR, adding a predictor (PR) to VICReg does not lead to a significant change of the performance, which indicates that PR is redundant with VR. In comparison, without VR, the representations collapse, and both stop-gradient (SG) and PR are necessary. Batch normalization in the inner layers of the expander (BN) in VICReg leads to a 1.0% increase in the performance, which is not a big improvement considering that SG and PR without BN is performing very poorly at 35.1%.

Table 3: **Effect of incorporating variance and covariance regularization in different methods.** Top-1 ImageNet accuracy with the linear evaluation protocol after 100 pretraining epochs. For all methods, pretraining follows the architecture, the optimization and the data augmentation protocol of the original method using our reimplementation. ME: Momentum Encoder. SG: stop-gradient. PR: predictor. BN: Batch normalization layers after input and inner linear layers in the expander. No Reg: No additional regularization. Var Reg: Variance regularization. Var/Cov Reg: Variance and Covariance regularization. Unmodified original setups are marked by a †.

| Method | ME | SG | PR | BN | No Reg | Var Reg | Var/Cov Reg |
|--------|----|----|----|----|--------|---------|-------------|
| BYOL | ✓ | ✓ | ✓ | ✓ | 69.3† | 70.2 | 69.5 |
| SimSiam | | ✓ | ✓ | ✓ | 67.9† | 68.1 | 67.6 |
| SimSiam | | ✓ | ✓ | | 35.1 | 67.3 | 67.1 |
| SimSiam | | ✓ | | | collapse | 56.8 | 66.1 |
| VICReg | | | ✓ | | collapse | 56.2 | 67.3 |
| VICReg | | | ✓ | ✓ | collapse | 57.1 | 68.7 |
| VICReg | | | | ✓ | collapse | 57.5 | 68.6† |
| VICReg | | | | | collapse | 56.5 | 67.4 |

Finally, incorporating VR with SG or ME further improves the performance by small margins of respectively 0.2% and 0.9%, which might be explained by the fact that these architectural tricks that prevent collapse are not perfectly maintaining the variance of the representations, i.e. very slow collapse is happening with these methods. We explain this intuition by studying the evolution of the standard deviation of the representations during pretraining for BYOL and SimSiam in Appendix D. We then analyse the impact of adding additional covariance regularization (CR) in the different settings, along with variance regularization. We found that optimization with SG and CR is hard, even if our analysis of the average correlation coefficient of the representations during pretraining in Appendix D shows that both fulfill the same objective. The performance of BYOL and SimSiam slightly drops compared to VR only, except when PR is removed, where SG becomes useless. BN is still useful and improves the performance by 1.3%. Finally with CR, PR does not harm the performance and even improves it by a very small margin. VICReg+PR with 1000 epochs of pretraining exactly matches the score of VICReg (73.2% on linear classification).

**Weight sharing.** Contrary to most self-supervised learning approaches based on Siamese architectures, VICReg has several unique properties: (1) weights do not need to be shared between the branches, each branch's weights are updated independently of the other branch's weights; (2) the branches are regularized independently, the variance and covariance terms are computed on each branch individually; (3) no predictor is necessary unlike with methods where one branch predicts outputs of the other branch. Table 4 shows results on ImageNet using the standard linear protocol for situations where the weights of the encoder and the expander are shared or not. In all settings, there is no collapse and the performance is competitive. The slight drop in accuracy without sharing is likely due to the increased number of parameters. Importantly, **the ability of VICReg to function with different parameters, architectures, and input modalities for the branches widens the applicability to joint-embedding SSL to many applications, including multi-modal signals.**

Table 4: **Impact of sharing weights or not between branches.** Top-1 accuracy on linear classification with 100 pretraining epochs. In all settings, the encoder and expander of both branches share the same architecture, but either share weights (✓), or have different weights in the two branches.

| Encoder | Expander | Top-1 |
|---------|----------|-------|
| | | 66.5 |
| | ✓ | 67.3 |
| ✓ | | 67.8 |
| ✓ | ✓ | 68.6 |

**Loss function coefficients.** Table 5 reports the performance for various values of the loss term coefficients in Eq. (6). Without variance regularization the representations immediately collapse to a single vector and the covariance term, which has no repulsive effect preventing collapse, has no impact. The invariance term is absolutely necessary and without it the network can not learn any good representations. By simply using the invariance term and variance regularization, which is a very simple baseline, VICReg still reaches an accuracy of 57.5%. These results show that variance and covariance regularizations have complementary effects.

Table 5: **Impact of variance-covariance regularization.** Inv: a invariance loss is used, $\lambda > 0$, Var: variance regularization, $\mu > 0$, Cov: covariance regularization, $\nu > 0$, in Eq. (6).

| Method | $\lambda$ | $\mu$ | $\nu$ | Top-1 |
|---|---|---|---|---|
| Inv | 1 | 0 | 0 | collapse |
| Inv + Cov | 25 | 0 | 1 | collapse |
| Inv + Cov | 0 | 25 | 1 | collapse |
| Inv + Var | 1 | 1 | 0 | 57.5 |
| Inv + Var + Cov (VICReg) | 25 | 25 | 1 | 68.6 |

Table 6: **Impact of normalization.** Std: variables are centered and divided by their standard deviation over the batch. This is applied or not to the embedding and the expander hidden layers. $l_2$: the embedding vectors are $l_2$-normalized.

| Expander | Embedding | Top-1 |
|---|---|---|
| Std | None | 68.6 |
| Std | Std | 68.4 |
| None | None | 67.4 |
| None | Std | 67.2 |
| Std | $l_2$ | 65.1 |

**Normalizations.** VICReg is the first self-supervised method for joint-embedding architectures we are aware of that does not require normalization. Contrary to SimSiam, W-MSE, SwAV and BYOL, and others, the embedding vectors are not projected on the unit sphere. Contrary to Barlow Twins, they are not standardized (equivalent to batch normalization without the adaptive parameters). Table 6 shows that the best settings do not involve any normalization of the embeddings, whether it is batch-wise or feature-wise (as in $l_2$ normalization). Whenever the embeddings are standardized (lines 3 and 5 in the table) the covariance matrix of Eq. (3) becomes the normalized auto-correlation matrix with coefficients between -1 and 1. This hurts the accuracy by 1.1%. We observe that when unconstrained, the coefficients in the covariance matrix take values in a wider range, which seems to facilitate the training process. Standardization is still an important component that helps stabilize the training when used in the hidden layers of the expander, and the performance drops by 1.2% when it is removed. Projecting the embeddings on the unit sphere implicitly constrains their standard deviation along the batch dimension to be $1/\sqrt{d}$, where $d$ is the dimension of the vectors. We change the invariance term of Eq. (5) to be the mean square error between $l_2$-normalized vectors, and the target $\gamma$ in the variance term of Eq. (1) is set to $1/\sqrt{d}$ instead of 1, forcing the standard deviation to get closer to $1/\sqrt{d}$, and the vectors to be spread out on the unit sphere. This puts a lot more constraints on the network and the performance drops by 3.5%.

# 7 Discussion

We introduced VICReg, a simple approach to self-supervised learning based on a triple objective: learning invariance to different views with a invariance term, avoiding collapse of the representations with a variance preservation term, and maximizing the information content of the representation with a covariance regularization term. VICReg achieves results on par with the state of the art on many downstream tasks, but is not subject to the same limitations as most other methods, particularly because it does not require the embedding branches to be identical or even similar.

**Limitations.** The time and memory costs of VICReg are dominated by the computation of the covariance matrix for each processed batch, which is quadratic in the dimension of the embeddings. Our experimental analysis, which corroborates the analysis of [9], shows that increasing the dimension of the embeddings significantly improves performance. Future work will explore how this quadratic bottleneck can be overcome by different approximation techniques, as well as completely new information maximization approaches based on higher-order statistics, and whether large expander networks are required.

SwAV [5] introduced multi-crop, a data-augmentation protocol where more than two views are produced for each image, which improves considerably the performance on downstream tasks. Using multi-crop with VICReg did not yield any performance improvement and showed signs of overfitting. More generally, multi-crop does not seem to help VICReg, Barlow Twins [9], SimSiam [7] nor BYOL [6], but yields performance improvements with SwAV [5], SimCLR [12] and MoCo [3], which might be related to the fact that these methods are contrastive.

**Broader impact.** This work increases the domain of applicability of self-supervised learning, and may improve the performance on tasks for which labeled data is scarce, visual or otherwise, such as healthcare, environmental protection, material science, and the understanding and translation of rare languages. Using the method described here is not likely to mitigate the usual issues with machine-learning systems due to biases in the data or the model architecture.

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
