# OpenReview forum: "VICReg: Variance-Invariance-Covariance Regularization for Self-Supervised Learning"
_NeurIPS.cc/2021/Conference — NeurIPS 2021 Submitted_

### Official Review · Reviewer_oH4g · 2021-07-15

**Rating:** 5
**Confidence:** 4

**Summary:**

This paper proposes a Variance-Invariance-Covariance regularization (VICReg) for self-supervised learning. The total loss function is the weighted sum of three terms: 1. distance between two views of the same input batch; 2. hinge loss on the variance of the embedding variables, encouraging them to stay away from zero; 3. loss on the covariance between different embedding variables, encouraging them to be close to zero, and hence preventing a representation collapse. The proposed model does not require the embedding branches to be identical, nor does it require mining of contrastive pairs. Experiments are conducted to demonstrate the effectiveness of the proposed regularization.

**Limitations And Societal Impact:**

Yes.

**Main Review:**

** Pros **
1. The paper is well written and well organized. It is easy to read.
2. The proposed variance term is easy to understand and implement.
3. Experiments and ablation study has been conducted to verify the effectiveness of each component of the proposed VICReg.

** Cons **
1. The novelty and contribution of the proposed work seems to be limited: the authors have stressed multiple times that VICReg does not require the two branches to be identical. But, if I understand correctly, the prior work Barlow Twins also does not have such restriction.
2. Also, the work seems to be built on the idea of Barlow Twins, with the only addition of the variance term in the proposed VICReg.
3. The fact that the proposed work does not require batch-wise or feature-wise normalization does not strike me as extremely important. After all, in table 6, the accuracy drop after standardizing the embedding is only 0.2%
4. The choice of hyperparameter also seems a bit willy-nilly -- why is $\lambda = \mu$? Is there any specific reason for such choice?
5. The method has only comparable accuracy compared to Barlow Twins, which makes me wonder what is the major gain from using variance regularization.
6. In 5.1, the authors mentioned that the performance of VICReg is on par with state of the art without using asymmetric networks architectural tricks such as the momentum encoder of BYOL. Is there going to be performance gain when such asymmetric network tricks are used? After all, the fact that the proposed architecture does not need to be symmetric is a highlight of the work.

**Review after author response**
I appreciate author's response addressing my concerns (2-6). The preliminary experiment on comparing Barlow Twins and the proposed work is also promising, but more study on the performance gain is warranted. I am raising my rating to 5.

**Time Spent Reviewing:**

3

---

> ### Author Response · Authors · 2021-08-09
> **Novelty and experiments with architectures with different weights**
>
> * *The novelty and contribution of the proposed work seems to be limited: the authors have stressed multiple times that VICReg does not require the two branches to be identical. But, if I understand correctly, the prior work Barlow Twins also does not have such restriction.*
>
> We agree that there is no inherent architectural limit preventing Barlow
> Twins from using two network branches with different weights. To compare
> the robustness of VICReg and Barlow Twins to different weights and/or
> architectures in the two branches, we have conducted a preliminary
> experiment: We have tested three scenarios: SWSA (for Shared Weights for
> the encoders and expanders, and Same Architecture for the encoders, a
> Resnet-50 in this case),  DWSA (for Different Weights and Same
> Architecture for the encoders and expanders, again a Resnet-50), and DA
> (for different architectures ResNet-50 for the first branch and
> ResNet-101 for the second one, and of course different weights for the
> corresponding encoders and expanders). Accuracy results  on linear
> evaluation on ImageNet for these three scenarios are
> results are presented in the Table below.
>
> |                        | SWSA     | DWSA |    DA |
> | ---                   | ---         | ---           | ---      |
> | VICReg          | 68.6     | 66.5        | 68.1 |
> | Barlow Twins  |  68.7    | 61.4        | 60.2 |
>
>
> Although these results are preliminary, they suggest that VICReg is much
> more robust than Barlow Twins to the use of different weights or
> architectures in the two branches of the network. More experiments are
> needed to truly ascertain this property of course. A plausible
> explication for this phenomenon might lie in the fact that the
> cross-correlation regularizer used by Barlow Twins does not have (so
> far) a clear interpretation, in general and in this case in particular,
> whereas the variance/covarianche regularizers used by VICReg act on the
> two branches independently, preserving their clear role even in that case.
>
> * *Also, the work seems to be built on the idea of Barlow Twins, with the only addition of the variance term in the proposed VICReg.*
> * *The method has only comparable accuracy compared to Barlow Twins, which makes me wonder what is the major gain from using variance regularization.*
>
> The variance term is indeed the main contribution of our paper. It is a simple and efficient solution against the collapse problem in self-supervised learning, and can additionally be incorporated into other methods, increasing the stability and accelerating the convergence of the training. We refer to our answer to cVf4 for a complete explanation on the benefits of the variance term.
>
> * *The fact that the proposed work does not require batch-wise or feature-wise normalization does not strike me as extremely important.*
>
> Normalization is required for every other self-supervised methods. We believe that removing it keeps the method very simple, and facilitates a potential future theoretical analysis.
>
> * *The choice of hyperparameter also seems a bit willy-nilly.*
>
> We provide in our answer to a9mx an explanation of how we chose the hyper-parameter of our loss function. We believe that tunning these parameters is not very complicated, and that using the default values we provide transfers well to other datasets than ImageNet.
>
> * *In 5.1, the authors mentioned that the performance of VICReg is on par with state of the art without using asymmetric networks architectural tricks such as the momentum encoder of BYOL. Is there going to be performance gain when such asymmetric network tricks are used?*
>
> The main interests of having independent branches is the potential applicability to multi-modal setups where both branches process different type of data. We are currently running such experiments with VICReg, using audio and video as input. We detail these experiments in our answer to a9mx. In addition, our preliminary experiment show that when using two different encoders, the encoder with the lowest capacity benefits from the encoder with the biggest capacity, which leads to performance gain with asymmetric network, and is a promising direction for future work.

---

### Official Review · Reviewer_W5JX · 2021-07-16

**Rating:** 5
**Confidence:** 4

**Summary:**

The authors propose a self-supervision technique with a more flexible structure requirements such as  not requiring batch norm, normalization, weight sharing or quantization.  Their model uses 2 encoders and a loss function with 3 variance and covariance regularization terms.  The authors mention that compared with the previous works, their main novelty lies in one of their variance terms which adjusts the variance of each input dimensions and, in return expected to mitigate collapsing problem. The authors claim that not sharing weights and having regularization terms separate on each encoder branch make their model better suited for multimodal inputs and tasks. They show results mainly on ImageNet classification, object detection tasks, with an ablation study on their and other baseline methods by varying regularization terms.

**Limitations And Societal Impact:**

The authors did a great job in presenting their work's limitations, their results in general not being better than the previous works and their extensive analysis(tables). If they did a better job in explaining the reasons/intuitions in a more solid way, or include some theory if there is any, I would be inclined to give an accept.

**Main Review:**

##### *Originality:*

- The idea of trying to prevent collapsing and using decorrelations, also working on a more flexible model exists in the recent works. The authors' main contribution lies in the details of the formulation of their loss. For example, if we compare with Barlow twins method, we can say that authors' invariance term corresponds to changing the cosine metric to euclidean for the diagonal covariance terms(so the same idea but different metric here). They kept covariance regularization separate for each embedding model as another difference. Exactly using hinge loss to control standard deviation dimension-wise is novel as of my knowledge.
- Overall, the work is novel in the sense that they collected all these details in one model, but I see these modifications as training tricks rather than a new loss function for self supervision, due to a lack of any theoretical analysis or at least a proof of concept with simpler cases(My opinions about the intuition part are below).

- The  authors did a great job in the supplementary to compare and separate themselves from the previous works.(Also Figure 2 is very informative) I believe they could be mentioned in the main paper in a better, more organized way.

- The related work was adequate, figures and comparisons in the supplementary section could be a good part of a survey paper in my opinion.

- I appreciate the authors efforts for their detailed and transparent ablation analysis (Table 3,5,6 ), a good example for our field.

&nbsp;
##### *Quality:*

- The authors both present their weaknesses and strengths in their analysis with sufficient experimental detail, it is admirable, but they could provide more intuition why other methods do better than theirs.

- The claims could be better supported. Some examples and questions(if I did not miss out anything)
     - Why using normalization is a problem for a network or a task (it can be thought as a part of cosine distance)? How would Barlow Twins perform if their invariance term is replaced with a euclidean distance?

     - Your method still uses 2048 as the batch size, I would not consider it as small. For example, Simclr uses examples in the same batch and its batch size changes between 256-8192. Most of the methods you mentioned need even much lower batch size.

     - You mentioned not sharing weights as an advantage, but you have shared weights in your results, except Table 4 in which the results degraded as you mentioned. What stops the other methods from using different weights? It should be possible even though they have covariance term between the embeddings, how much their performance would be affected compared with yours?
My intuition is that a proper design might be sufficient rather than separating variance terms.

     - Do you have a demonstration or result related to your model collapsing less than other methods? In line 159, you mentioned gradients become 0 and collapse; it was a good point, is it commonly encountered, did you observe it in your experiments?

     - I am also not convinced to the idea that the images and their augmentations need to be treated separately, they can be interchangeable.

    - Variances of the results could be included to show the stability of the algorithms since it was another claim in the paper(although "collapsing" shows it partly, it is a biased criteria since the other methods are not designed for var/cov terms).

    - How hard is it to balance these 3 terms?

    - When someone thinks about gathering two batches from two networks and calculate the global batch covariance in this way; it includes both your terms and Barlow Twins terms. Can anything be said based on this observation, about which one is better and why?

&nbsp;

##### *Significance:*

- Currently, the paper needs more solid intuition or analysis  or better results to make an impact in my opinion. The changes compared with the prior work are minimal. Most of the ideas and problems in the paper are important, but they are already known.

- The comparisons with the previous work is valuable to the field, they could maybe extend their experiments to the more of the mentioned methods or other variants.




**Time Spent Reviewing:**

4

---

> ### Author Response · Authors · 2021-08-09
> **Normalization, Batch size and diversity of the representations**
>
> * *Why using normalization is a problem for a network or a task (it can be thought as a part of cosine distance)? How would Barlow Twins perform if their invariance term is replaced with a euclidean distance?*
>
> We believe that removing any non-essential tricks, such as normalization in this case, is a good thing for a learning architecture, keeping it as simple as possible. The performance of Barlow Twins drops from 71.4\% to 53.4\% when no normalization is used. Moreover the representations of Barlow Twins completely collapse when the l2-distance is used, which may be linked to the fact that we do not have a good explanation of why Barlow Twins representations do not collapse.  VICReg on the other hand explicitly prevents collapse by design of the loss function. We explain our hypothesis in our answer to cVf4.
>
> * *Your method still uses 2048 as the batch size, I would not consider it as small. For example, Simclr uses examples in the same batch and its batch size changes between 256-8192*
>
> The performances of VICReg and SimCLR drop by respectively 1.3\% and 2.6\% when a batch size of 256 is used, and by 4.5\% and 12.8\% when a batch size of 64 is used. This shows that VICReg is much more robust than SimCLR to the size of the batch. We will provide additional experiments with batch size $<$ 256 in the final manuscript.
>
> * *You mentioned not sharing weights as an advantage, but you have shared weights in your results, except Table 4 in which the results degraded as you mentioned*
>
> The main advantage of an architecture with branches with different weights is not that it  yields a performance improvement (which may or may not be the case), but it is the ability and potential to process multi-modal data using encoders with very different architectures.
>
> * *What stops the other methods from using different weights? What stops the other methods from using different weights? It should be possible even though they have covariance term between the embeddings, how much their performance would be affected compared with yours?*
>
> The methods based on a teacher/student setup where the teacher's weights are a moving average of the student's weights, such as BYOL, MoCo and OBoW, cannot by construction have encoders for the two branches with different architectures. More generally, the methods that do not have a symmetric architecture, such as SimSiam and SwAV, cannot be used in multi-modal setups. Among non-contrastive self-supervised methods, Barlow Twins is the only one which might be used in these setups. Our preliminary experiments presented in our answer to oH4g shows that Barlow Twins severely under-performs when two encoders that do not share weights are used, and that VICReg is more robust in these setups.
>
> * *Do you have a demonstration or result related to your model collapsing less than other methods? In line 159, you mentioned gradients become 0 and collapse; it was a good point, is it commonly encountered, did you observe it in your experiments?*
>
> When using the variance directly instead of the standard deviation in the variance term of VICReg's loss function, we empirically observed the collapse explained in l159. This collapse is completely avoided by using the standard deviation.
>
> * *I am also not convinced to the idea that the images and their augmentations need to be treated separately, they can be interchangeable.*
>
> All the recent self-supervised methods based on siamese networks use the same data augmentation protocol, which consists in comparing two augmented views of an image. VICReg simply follows standard practice and uses the same set of augmentation as these methods.
>
> * *How hard is it to balance these 3 terms?*
>
> We refer to the explanation given in our answer to a9mx on how we balance the 3 terms of our loss function.
>
> * *Variances of the results could be included to show the stability of the algorithms since it was another claim in the paper.*
>
> We provide in our answer to cVf4 an explanation of the benefice of incorporating VReg, the variance part of our loss function, in other self-supervised methods. VReg increases the stability of the training and accelerates convergence. We also provide in the following table the final standard deviation values of the representations after projection on the unit sphere, averaged on the validation images of ImageNet, for different methods. This metric was used in our initial experiments with VICReg, to measure how well the representations are spread out on the unit sphere (when projected on it), and therefore is a useful indication on how diverse the representations are.
>
> | VICReg         | Barlow Twins  | BYOL          | SimSiam        |
> | --- | --- | --- | --- |
> | 0.01885 | 0.01805 | 0.01674 | 0.01745 |
>
> These results show that VICReg produces the most diverse set of representations. We will include them in the final manuscript, as well as the evolution of this standard deviation metric during training for different methods.
>
> * *When someone thinks about gathering two batches from two networks and calculate the global batch covariance in this way; it includes both your terms and Barlow Twins terms. Can anything be said based on this observation, about which one is better and why?*
>
> Contrary to Barlow Twins, the covariance term of VICReg is calculated independently on the two branches. This allows for more flexibility, and for the branches to have different architectures and process different input data modalities.

---

### Official Review · Reviewer_a9mx · 2021-07-16

**Rating:** 8
**Confidence:** 5

**Summary:**

Summary:
This paper proposes a new self-supervised learning method. It makes self-supervised learning very elegant: it does not require techniques such as weight sharing between the branches, batch normalization, feature-wise normalization, output quantization, stop gradient, memory banks, etc., and achieves results on par with the state-of-the-art results on several downstream tasks. It does not require that the inputs be of the same nature. This opens the door to the use of non-contrastive self-supervised joint-embedding for multi-modal signals, such as video and audio. I really like this paper. Still, I have some concerns about this paper.


**Ethical Concerns:**


No.


**Limitations And Societal Impact:**



Please refer to and respond to the "negative" items in the above main review.



**Main Review:**



(Positive) This paper correctly understands the critical problem in self-supervised learning, i.e., the main challenge is to prevent a collapse in which the encoders produce constant or non-informative vectors.

(Positive) This paper makes self-supervised learning very elegant: it does not require techniques such as weight sharing between the branches, batch normalization, feature-wise normalization, output quantization, stop gradient, memory banks, etc., and achieves results on par with the state-of-the-art results on several downstream tasks.

(Positive) This paper correctly understands the critical problem in BYOL, i.e., the dynamics of learning and how they avoid collapse, is not fully understood.

(Negative) There is a tradeoff for Variance, Invariance, and Covariance. I am afraid that different hyper-parameter will result in very different performances. If so, the proposed method will not be robust. Especially when I see Coefficients λ and μ are 25. Could the authors provide some dense results/visualization curves beyond Table 5?

(Positive) The following property is fascinating: it does not require that the inputs be of the same nature. This opens the door to the use of non-contrastive self-supervised joint-embedding for multi-modal signals, such as video and audio. I really like this property very much!
Importantly, this good property also indicates: The ability of VICReg to function with different parameters, architectures, and input modalities for the branches widens the applicability of joint-embedding SSL to many applications, including multi-modal signals.

(Negative) Unfortunately, in most of our experiments, this paper uses a Siamese net architecture in which the two branches are identical and share weights.

(Negative) Although I like the above property very much, finding a real-world case meeting the property is difficult.  Because if we cross-modal learning, this is supervised learning but NOT unsupervised learning. Please refer to OpenAI's CILP. Could the authors provide a real-world case meeting the above property with an experimental result?

(Negative) In the overall paper, the authors consider BYOL and MOCO very different. Specifically, BYOL is regarded as a distillation method, while MoCo is considered as a contrastive learning method. I am not convinced by this categorization. If the authors think deeply, they will find that BYOL and MoCo are almost the same, except that MoCo has negative samples. The EMA networks (or mean teachers)  in MoCo and BYOL are exactly the same. If BYOL is categorized into knowledge distillation, MoCo could also be considered knowledge distillation, distilling knowledge from both positive and negative samples. Actually, I tend not to categorize methods like BYOL into knowledge distillation because the teacher is just a mean teacher with dynamic changes during the training. Specifically, SimSiam even has no mean teacher, and the learning target is just a copy of the student networks without BP.

(Negative) Is the following sentence correctly written?
Line 107-108: These methods are effective, but there is no clear understanding of how t but suffer from a lack of explainability regarding the way collapsing solutions are avoided.

 (Neutral) I think this paper may over-attribute some contributions from Barlow Twins. Actually, it borrows the decorrelation mechanism of the Barlow Twins. But it includes an explicit variance-preservation term for each variable of the two embeddings and thus does not require any normalization.

(Positive) The description of both motivation and the method are clear and easy to follow.

(Negative) Would you please provide an empirical analysis of gamma in Eqn (1), which is a constant target value for the standard deviation?

(Positive) The following observation is good:
"Using the standard deviation and not directly the variance is crucial. Indeed, if we take S(x) = Var(x) in the hinge function, the gradient of S with respect to x becomes close to 0 when x is close to x ̄. In this case, the gradient of v also becomes close to 0, and the embeddings collapse."

(Negative) The following results are not good: "We report the performance in Table 2, VICReg performs on par with most concurrent methods, and better than Barlow Twins, across all classification tasks, but is slightly behind the top-3 on detection tasks. "
The explanation is not persuasive: "This could be explained by the fact that VICReg learns representations that are more invariant to transformation, but eliminates more low-level information about the images than the other methods."

(Neutral) According to Table 3, the proposed method should have used symmetric loss (i.e., Eqn. (8)). Please clarify.

(Negative) The following sentence is confusing:
Line 315: This hurts the accuracy by 1.1%. Who is compared with who?

(Negative) Two online networks increase both the memory and the running time.

(Positive) I strongly agree with the following observation (in my experiments, I also have these observations): "Using multi-crop with VICReg did not yield any performance improvement and showed signs of overfitting. More generally, multi-crop does not seem to help VICReg, Barlow Twins [9], SimSiam [7] nor BYOL [6], but yields performance improvements with SwAV [5], SimCLR [12] and MoCo [3], which might be related to the fact that these methods are contrastive."

(Negative) Provide evidence for the following claim: "Self-supervised learning is also trained on data, which may also have a bias.
Using the method described here is not likely to mitigate the usual issues with machine-learning systems due to biases in the data or the model architecture."

(Negative) The authors claim that multi-cropping yields performance improvements with SwAV [5], SimCLR [12] and MoCo [3], which might be related to the fact that these methods are contrastive. This is not true. For example, DINO and the Triplet loss method are not in line with this claim. They don't belong to contrastive learning, but Multi-crop learning does help them. Maybe the gain is related to loss function but not contrastive learning.
DINO: Emerging Properties in Self-Supervised Vision Transformers
Triplet Loss: Towards Solving Inefficiency of Self-supervised Representation Learning

(Negative) Because the proposed method can handle two different networks with different architectures and different weights. I would like to see such an example. Especially when one network is ResNet-50 and another network is a larger network than ResNet-50 (e.g., ResNet-101 or a strong ViT). Could the linear evaluation result of ResNet-50 be very high in these cases? It would be interesting to see such a result because it implicitly contains a knowledge distillation effect that distills the knowledge from the larger network.


**Time Spent Reviewing:**

24

---

> ### Author Response · Authors · 2021-08-09
> **Hyper-parameters tuning and experiments with multi-modal data**
>
> * *I am afraid that different hyper-parameter will result in very different performances. If so, the proposed method will not be robust*
>
> Let us clarify how the values of the hyper-parameters of VICReg have
> been chosen: First, we have empirically found that using very different
> values for lambda and mu, or taking lambda=mu with nu>mu leads to
> unstable training. On the other hand taking lambda=mu and picking nu<mu
> leads to stable convergence, with the exact value picked for mu having
> very limited influence on the final linear classification accuracy. This
> is illustrated by the table below, showing accuracy after 100 epochs of
> training.
>
> | lambda | mu  | nu      | accuracy 100 epochs |
> | ---     | --- | ---     | ---      |
> | 1       | 1   | 1       | Collapse |
> | 1       | 10  | 1       | Collapse |
> | 10      | 1   | 1       | Collapse |
> | 5       | 5   | 1       | 68.1 |
> | 10      | 10  | 1       | 68.2 |
> | 25      | 25  | 1       | 68.6 |
> | 50      | 50  | 1       | 68.3 |
>
> We have found that setting  lambda=mu= 25 and nu=1 works best (by a small margin) for
> Imagenet but we have also obtained excellent results
> on MNIST and Cifar-10 and 100 using these exact same values. We could
> easily have tuned these parameters by cross-validation on the validation
> sets of these two smaller datasets.
>
> * *Unfortunately, in most of our experiments, this paper uses a Siamese net architecture in which the two branches are identical and share weights.*
>
> We agree that more experiments with architectures with branches that do not share weights  are required. We provide in our answer to oH4g the results of a preliminary experiment where two different architectures are used for both branches, that shows that VICReg is very robust in these settings.
>
> * *Although I like the above property very much, finding a real-world case meeting the property is difficult. Because if we cross-modal learning, this is supervised learning but NOT unsupervised learning. Please refer to OpenAI's CILP. Could the authors provide a real-world case meeting the above property with an experimental result?*
>
> Cross-modal learning does not necessarily require supervision. For example, we are currently conducting the following experiments with VICReg: 1) We use videos in one branch and the corresponding audio signals in the other to learn meaningful video representations. 2) We use videos to learn a common representation between a few consecutive frames in one branch, encoded with a video encoder, and a frame from the future in the other branch, encoded with a frame encoder. In both cases VICReg is used to model the dependencies between the two inputs, and works with different encoder architectures in both branches, and different data modalities.
>
> * *In the overall paper, the authors consider BYOL and MOCO very different. Specifically, BYOL is regarded as a distillation method, while MoCo is considered as a contrastive learning method. I am not convinced by this categorization.*
>
> We agree that MoCo is also a teacher/student based method and we will clarify this point in our related work section.
>
> * *Is the following sentence correctly written? Line 107-108: These methods are effective, but there is no clear understanding of how t but suffer from a lack of explainability regarding the way collapsing solutions are avoided*
>
> We will rephrase as "These methods are effective, but there is no clear understanding of why and how they avoid collapse"
>
> * *Would you please provide an empirical analysis of gamma in Eqn (1)*
>
> We take gamma=1 for simplicity, but we found that using 0.1 < gamma < 10.0 works just as well. Using values of gamma outside of this range leads to instabilities and requires additional tuning of the loss hyper-parameters.
>
> * *The explanation is not persuasive: "This could be explained by the fact that VICReg learns representations that are more invariant to transformation, but eliminates more low-level information about the images than the other methods."*
>
> This explanation is only a hypothesis (and we do not claim it is anything else). Validating this hypothesis will require experiments to measure how invariant the representations from VICReg and Barlow Twins are to data augmentation. This is left for future work. We will clarify this point in the presentation.
>
> * *According to Table 3, the proposed method should have used symmetric loss (i.e., Eqn. (8)). Please clarify*
>
> We have chosen to keep the method simple and to not use a predictor network. We have found that with 1000 epochs, instead of 100 in the ablations of Table 3, using a predictor leads to the same performance as not using it.
>
> * *The following sentence is confusing: Line 315: This hurts the accuracy by 1.1\%. Who is compared with who?*
>
> The 1.1% accuracy drop was a typo. The actual drops is only 0.2% and corresponds to standardizing the embeddings or not. We will clarify this.
>
> * *Two online networks increase both the memory and the running time*
>
> We agree that using two online networks increases the memory usage. However, the running time do not increases significantly. Indeed, for each input, two forward passes must be completed in both cases. VICReg applied to only images does not require two online networks, but dealing with multiple modalities will imply the usage of two online networks and therefore a additional memory requirements.
>
> * *Provide evidence for the following claim: "Self-supervised learning is also trained on data, which may also have a bias. Using the method described here is not likely to mitigate the usual issues with machine-learning systems due to biases in the data or the model architecture."*
>
> We believe that the property of replicating data biases is shared by all current learning methods, deep or not, including all recent self-supervised methods. If no explicit mechanism is implemented, the algorithm will simply learn the biases in the data and reproduce them.
>
> * *The authors claim that multi-cropping yields performance improvements with SwAV [5], SimCLR [12] and MoCo [3], which might be related to the fact that these methods are contrastive. This is not true*
>
> We were not aware that multi-cropping does not yield performance improvements with the triplet loss, and the relation to contrastive learning was just an hypothesis. We therefore hypothesize that multi-cropping yields performance improvement in combination with the InfoNCE and the cross-entropy loss.
>
> * *Because the proposed method can handle two different networks with different architectures and different weights. I would like to see such an example*
>
> We provide in our answer to oH4g an additional experiment where the two encoders for the two branches consist in a ResNet-50 and a ResNet-101. Compared to the setup with two ResNet-50, the performance increases significantly, which validate the hypothesis that using a larger architecture for the other branch can benefit the performance of the ResNet-50 branch. Future work will explore this idea in more depth.

---

> > ### Comment · Reviewer_a9mx · 2021-08-22
> > **Thank you, reviewers and authors!**
> >
> >
> > Thank you, reviewers and authors!
> >
> > The author has answered almost all my questions. The reply is in-depth, especially in the following points (although there are some minor flaws).
> >
> > (Positive) The response to "how the values of the hyper-parameters of VICReg have been chosen" is very clear and very convincing.
> > -- "First, we have empirically found that using very different values for lambda and mu, or taking lambda=mu with nu>mu leads to unstable training. On the other hand, taking lambda=mu and picking nu<mu leads to stable convergence, with the exact value picked for mu having very limited influence on the final linear classification accuracy. "
> >
> > (Positive and negative) The authors' response to cross-modal learning is good but still, have some problem.
> > --"Cross-modal learning does not necessarily require supervision. For example, we are currently conducting the following experiments with VICReg: 1) We use videos in one branch and the corresponding audio signals in the other to learn meaningful video representations. 2) We use videos to learn a common representation between a few consecutive frames in one branch, encoded with a video encoder, and a frame from the future in the other branch, encoded with a frame encoder. "
> >
> > However, I don't think the audio can help the video representation learning. I also don't think image-to-video contrastive learning using separate networks can outperform video-to-video contrastive learning using shared networks.
> >
> > (Positive) The analysis of gamma in Eqn (1)  is very clear and very convincing.
> > --"We take gamma=1 for simplicity, but we found that using 0.1 < gamma < 10.0 works just as well. Using values of gamma outside of this range leads to instabilities and requires additional tuning of the loss hyper-parameters."
> >
> > (Neutral) By saying symmetric loss, I mean the two views (X and X') of an Image are feed into networks A and B, respectively, to get the loss(X_A, X'_B). Then, X and X' are feed into the networks B and A, respectively, to get the loss(X_B, X'_A). Then, there two losses are combined.
> >
> > (Positive) The authors' response to "the proposed method can handle two different networks with different architectures and different weights" is excellent and convincing.
> >
> > I have read all reviewers' comments and the authors' responses. I would like to provide a few remarks to the reviewers with my humble knowledge.
> >
> > To Reviewer #cVf4:
> > I really thank Reviewer #cVf4 for his/her treasurable comments! I understand that Reviewer cVf4 is concerned about the novelty of this paper and that the improvement of Variance Reg and covariance Reg on other self-supervised methods is marginal. Actually, I also think that this paper may over-attribute some contributions from Barlow Twins. It borrows the decorrelation mechanism of the Barlow Twins. But it includes an explicit variance-preservation term for each variable of the two embeddings and thus does not require any normalization. Although the performance gain is marginal, I think it can inspire many future works. Mainly if this method can be applied to cross-modal data (where the two networks receive data of different modalities), the value of this method will be very significant. I think this is a critical step, and this paper deserves acceptance.
> >
> > To Reviewer #W5JX:
> > I really thank Reviewer #W5JX for raising his/concerns and for his valuable comments! I understand that Reviewer #W5JX has some concerns about the current self-supervised learning methods and the proposed method. Because I have extensive experience in self-supervised learning experiments, I sincerely feel that the contribution of this article has indeed solved a critical problem of self-supervised learning. The reviewer believes, "My intuition is that a proper design might be sufficient rather than separating variance terms ..." In reality, very interestingly, a simple design might not solve this problem. The deeper we conduct experience in self-supervised learning, the more magics we will discover. I think some existing articles have answered some of the reviewer’s questions like "Do you have a demonstration or result related to your model collapsing less than other methods?" "Why using normalization is a problem for a network?" "What/why stops the other methods from using different weights?". For example, the article "Exploring Simple Siamese Representation Learning" discusses many phenomena related to the reviewers' questions. Actually, these questions are not easy to answer in theory, and we can only provide some empirical analysis so far. These questions deserve a comprehensive understanding in future work. In summary,  I think this is a crucial step, and this paper deserves acceptance.
> >
> >
> > To Reviewer #oH4g
> > I really appreciate Reviewer #oH4g's concern that "if I understand correctly, the prior work Barlow Twins also does not have such restriction." This is a very critical comment. I think the authors' response might have addressed this concern partially.  The reviewer also has a concern about the incremental contribution compared to Barlow Twins. Actually, I also think that this paper may over-attribute some contributions from Barlow Twins. It borrows the decorrelation mechanism of the Barlow Twins. But it includes an explicit variance-preservation term for each variable of the two embeddings and thus does not require any normalization. Although the performance gain is marginal, I think it can inspire many future works. Mainly if this method can be applied to cross-modal data (where the two networks receive data of different modalities), the value of this method will be very significant.
> >
> > In summary, because the author has answered almost all my questions, I would like to raise my rating from 7 to 8. Aslo, I would like to note the author that the suggestions from the other reviewers will undoubtedly benefit the paper's improvement and make the paper attract a wide range of audiences.

---

### Official Review · Reviewer_cVf4 · 2021-07-17

**Rating:** 5
**Confidence:** 3

**Summary:**

This paper proposes a simple and effective method(VICReg) that prevents the collapse in self-supervised learning with two regularizations terms. The variance term maintains the variance of the embedded vector along each dimension independently. The covariance term prevents the network encode similar information to the same dimension in the embedded space.

**Limitations And Societal Impact:**

Yes

**Main Review:**

Pros:
VICReg achieves comparable results with the SOTA on several downstream tasks.
VICReg does not require some training constraints on the architecture, which is more applicable.
The paper is clearly written and well organized.

Cons:
The covariance term is borrowed from the Barlow Twins method, which decreases the originality and significance.
The improvement of Variance Reg and covariance Reg on other self-supervised methods is marginal. It would be better if VICReg can be plugged into different methods.

**Time Spent Reviewing:**

12hours

---

> ### Author Response · Authors · 2021-08-09
> **Main differences with Barlow Twins and advantages of variance regularization**
>
>
> * *The covariance term is borrowed from the Barlow Twins method.*
>
> Contrary to Barlow Twins, VICReg explicitly prevents the collapse of the learned representation by construction: The variance term explictly prevents the different data points in the same batch from  admitting the same representation, and the covariance term prevents an informational collapse, that is when the information is concentrated in few dimensions of the representations. In contrast, although Barlow Twins empirically prevents collapse in practice, we are not aware of any heuristic or formal justification for that fact, in the original paper or elsewhere. Indeed, there is nothing preventing a priori Barlow Twins embeddings from shrinking to zero before normalization (the fact that this does not happen  might be linked to the small numerical constant added to the denominator of the batch norm layer used for normalization, but this would have to be ascertained). In contrast, the VICReg loss offers a very simple and explicit method for avoiding collapse, without the need for any trick or heuristic.
>
> The Barlow Twins loss computes a cross-correlation between the two branches, which is hard to interpret when the branches have very different statistics, for example when the two branches do not share their weights, do not share the same architecture or process different types of data. We provide a preliminary analysis in our answer to oH4g that validates these claims.
>
> * The improvement of Variance Reg and covariance Reg on other self-supervised methods is marginal. It would be better if VICReg can be plugged into different methods.
>
> VReg, the variance part of VICReg, can indeed easily be incorporated
> into other methods, leading in general to better stability and faster
> convergence during training. This is in fact clearly demonstrated by our
> experiments in Appendix C.5, where we incorporate VReg in BYOL and
> SimSiam. The final performance gain at test time is
> marginal because, even without VReg, these two methods eventually
> converge to solutions avoiding collapse, but at the cost of more epochs.
> See also our answer to W5JX showing that VICReg
> produces in general more "diverse" (high-variance) embeddings than competing methods.

---

### Decision · Program_Chairs · 2021-09-27

**Decision:**

Reject

**Comment:**

3 of the 4 reviewers voted to reject this paper based on the lack of significant differences from prior work, especially Barlow Twins, and the lack of SOTA numerical results. One concern I share with a number of the reviewers is that what is "simple" is somewhat subjective. E.g. it is not obvious that normalization is especially complex or replacing it with a (co)-variance regularizer is necessarily "simpler". While one reviewer points to potential significant gains with multi-modal data, this is a hypothesis not supported by specific claims in the paper as it stands, and therefore that cannot be considered in the current review process. As a result, I am recommending rejection. However, the approach will be a good paper with some revisions (e.g. additional results on multi-modal data).